# Alkaline Hydrothermal Treatment of Chabazite to Enhance Its Ammonium Removal and Recovery Capabilities through Recrystallization

**Dipshika Das and Sukalyan Sengupta ***

Civil & Environmental Engineering Department, University of Massachusetts Dartmouth,
North Dartmouth, MA 02747-2300, USA; ddas@umassd.edu
* Correspondence: ssengupta@umassd.edu

**Abstract:** The treatment of chabazite (CHA), a natural zeolite, with the alkaline hydrothermal method to improve its ion-exchange capacity is a widely adopted route by environmental scientists for the purpose of better ammonium ($NH_4^+$) removal from wastewater. This work addresses a noteworthy trend in environmental science, where researchers, impressed by the increased ion-exchange capacity achieved through alkaline hydrothermal treatment, often bypass the thorough material characterization of treated CHA. The prevalent misconception attributes the improved features solely to the parent zeolitic framework, neglecting the fact that corrosive treatments like this can induce significant alterations in the framework and those must be identified with correct nomenclature. In this work, alkaline-mediated hydrothermally treated CHA has been characterized through X-ray powder diffraction (XRD), Fourier transform infrared spectroscopy (FTIR), solid-state magic-angle spinning nuclear magnetic resonance (MAS-NMR), high-resolution transmission electron microscopy (HRTEM), and energy-dispersive X-ray spectroscopy (EDS) and it is concluded that the treated samples have been transformed into a desilicated, aluminum (Al)-dense framework of analcime (ANA) with a low silica–alumina ratio and with a strikingly different crystal shape than that of parent CHA. This treated sample is further examined for its $NH_4^+$ removal capacity from synthetic wastewater in a fixed-bed column arrangement. It achieved a maximum $NH_4^+$ removal efficiency of 4.19 meq/g (75.6 mg/g of $NH_4^+$), twice that of the parent CHA. Moreover, the regeneration of the exhausted column yielded a regenerant solution, with 94% reclaimed $NH_4^+$ in it, which could be used independently as a nitrogenous fertilizer. In this work, the meticulous compositional study of zeolitic materials, a well-established practice in the field of material science, is advocated for adoption by environmental chemists. By embracing this approach, environmental scientists can enhance their comprehension of the intricate changes induced by corrosive treatments, thereby contributing to a more nuanced understanding of zeolitic behavior in environmental contexts.

**Keywords:** chabazite (CHA); analcime (ANA); alkaline hydrothermal treatment; $NH_4^+$ removal; regeneration

## 1. Introduction

Natural zeolite is a type of aluminosilicate mineral with a unique crystalline structure, which comprises tetrahedral of silica ($SiO_4^{4-}$) and alumina ($AlO_4^{5-}$) linked by oxygen atoms. Each of these tetrahedrons has a negative charge that gets neutralized by alkali or alkaline earth metal cations. These cations can get replaced by external cations to which zeolite has a stronger affinity. This mechanism is known as ion exchange [1]. One such versatile natural zeolite is Chabazite (CHA). It finds application across various industries, including catalytic cracking and gas separation [2–5]. Due to its adsorption and ion-exchange properties, this material has been applied in water and wastewater treatment processes [6–8]. Its cation exchange capacity (CEC) positions it as a favorable option for

ammonium $(NH_4^+)$ removal from wastewater and agricultural runoff. The excessive use of nitrogen (N)-rich fertilizer on farmland leads to agricultural runoff contaminated with elevated levels of $NH_4^+$. Among many natural zeolites, CHA has excelled in terms of $NH_4^+$ removal efficiency through the ion-exchange process [9–13]. Moreover, researchers have observed that, with suitable regenerant composition or through calcination, exhausted CHA, loaded with removed contaminants, can be regenerated in simple steps and can be reintroduced in the removal cycle. This enables its use in a cyclical process in which they selectively remove targeted cations in sorption mode and undergo regeneration upon saturation, allowing for the removal of the targeted cations into a concentrated solution. The concentrated cation can be repurposed, such as $NH_4^+$ as a component of fertilizer; the zeolite can be reused in subsequent sorption cycles, substantially reducing treatment costs [9,10,14,15].

Scholars have explored different routes with the goal of improving key features of natural zeolite. There are primarily two methods, (i) a bottom-up approach and (ii) a top-down approach. As per the bottom-up approach, the zeolites can be tailor made with the desired silicon (Si)–aluminum (Al) ratio, mesoporosity, and with a molecular sieve property from fly ash [16,17], clay minerals [18], volcanic glasses [19] and aluminosilicate gel [20], etc. This process is accessorized with soft-templating or hard-templating chemicals to attain the desired feature. In the top-down approach, zeolite is improved through acid leaching or an alkaline hydrothermal treatment method [21]. During this approach, the intermittent processes of demetallation and recrystallization in the parent zeolite result in significantly enhanced characteristics in the treated zeolite [21–23]. Alkaline hydrothermal treatment is a well-established technique for enhancing the properties of natural zeolite. It provides an environment conducive to the conversion of an amorphous aluminosilicate phase into a stable zeolitic framework [24,25]. Due to the innate sensitivity of the lattices within natural zeolites, this treatment can induce changes in the original framework of the zeolite, facilitating recrystallization during the alkaline hydrothermal process. Additionally, this treatment results in the significant loading of cations in the extra framework of the original zeolite, ultimately leading to improved ion exchange and selectivity performance [13]. Multiple water chemists and environmental scientists have investigated the alkaline hydrothermal treatment of zeolite, such as clinoptilolite, CHA, etc. with the objective of enhancing its CEC for the efficient removal of $NH_4^+$ [9,10,26–28]. Nevertheless, these and other recent similar studies in this subject predominantly concentrate on enhancing CEC and tend to overlook the detailed characterization of the modified CHA. The increased CEC and other highlighted performance improvements are frequently attributed solely to the parent framework. However, there exists a reasonable likelihood of the reorientation of the parent zeolitic framework, resulting in a distinctly different framework.

In this work, CHA has been modified through the alkaline hydrothermal treatment method. Prior to the discussion of the performance of the modified CHA for $NH_4^+$ removal, this research elucidates the transformations undergone by CHA following alkaline hydrothermal treatment. Analytical methodologies, such as X-ray powder diffraction (XRD), Fourier transform infrared spectroscopy (FTIR), solid-state magic-angle spinning nuclear magnetic resonance (MAS-NMR), high-resolution transmission electron microscopy (HRTEM), and energy-dispersive X-ray spectroscopy (EDS), are employed to illustrate the conversion of CHA into another zeolite framework, analcime (ANA). This work substantiates the imperative need for a comprehensive material analysis of the enhanced zeolite modified through corrosive techniques, such as alkaline hydrothermal treatment, to ensure correct identification of the framework and further attribute the better performance as a facet of that. The efficiency of this modified CHA is further examined in a packed bed column to remove $NH_4^+$ from synthetic wastewater. It has shown selectivity toward $NH_4^+$ ions among the array of multiple cations in synthetic wastewater. The exhausted sample has undergone brine regeneration that yields a solution with considerable reclaimed $NH_4^+$ in it. With the addition of potassium and phosphorus in this solution, it may be an attractive starter for a commercial fertilizer.

## 2. Materials and Methods

Arizona Lower Bowie (AZLB)-Sodium (Na)-loaded Natural Chabazite (CHA), AZLB-Na CHA, was sourced from St. Cloud Mining (Bowie, AZ, USA). The material was washed multiple times in distilled water to eliminate zeolitic debris and salts and oven-dried at 55 °C. Dry AZLB-Na CHA was then sieved and granules > 850 μm were further used for the rest of the work. Aliquots of that AZLB-Na CHA were subjected to alkaline hydrothermal treatment in a closed reflux environment.

3 sets of 1 L solutions of 1.5 M, 2.5 M, and 3.5 M NaOH were prepared by addition of required mass of NaOH pellets. Those 9 solutions were used for the alkaline hydrothermal treatment optimization study in the following ways:

i. Twenty-five g of washed, cleaned, and dried AZLB-Na CHA in each of three 1 L solutions of 1.5 M NaOH; treated at 70 °C, 80 °C and 90 °C, respectively, for 24 h period;

ii. Twenty-five g of washed, cleaned, and dried AZLB-Na CHA in each of three 1 L solutions of 2.5 M NaOH; treated at 70 °C, 80 °C and 90 °C, respectively, for 24 h period;

iii. Twenty-five g of washed, cleaned, and dried AZLB-Na CHA in each of three 1 L solutions of 3.5 M NaOH; treated at 70 °C, 80 °C and 90 °C, respectively, for 24 h period.

Following the determination of the optimized NaOH concentration and solution temperature, seven separate alkaline hydrothermal treatments were carried out with two of those consistent parameters, wherein 25 g aliquots of AZLB-Na CHA were treated in a reflux environment with 1 L of alkaline solution for a progressively increasing number of days, 1, 2, 3, 4, 5, 6, and 7 d. The entire scheme is illustrated in Figure 1. Those last seven treated samples are examined further to conclude the extent of alkaline hydrothermal treatment. The treated AZLB-Na CHA, modified through optimized treatment, was washed multiple times with distilled and deionized water, and the pH was neutralized using HCl. The sample was oven-dried at 55 °C. This sample was labeled as follows: AZLB-Na CHA treated in a y M NaOH solution at x °C for z d was denoted as AHTCHA$^{zDxCyM}$.

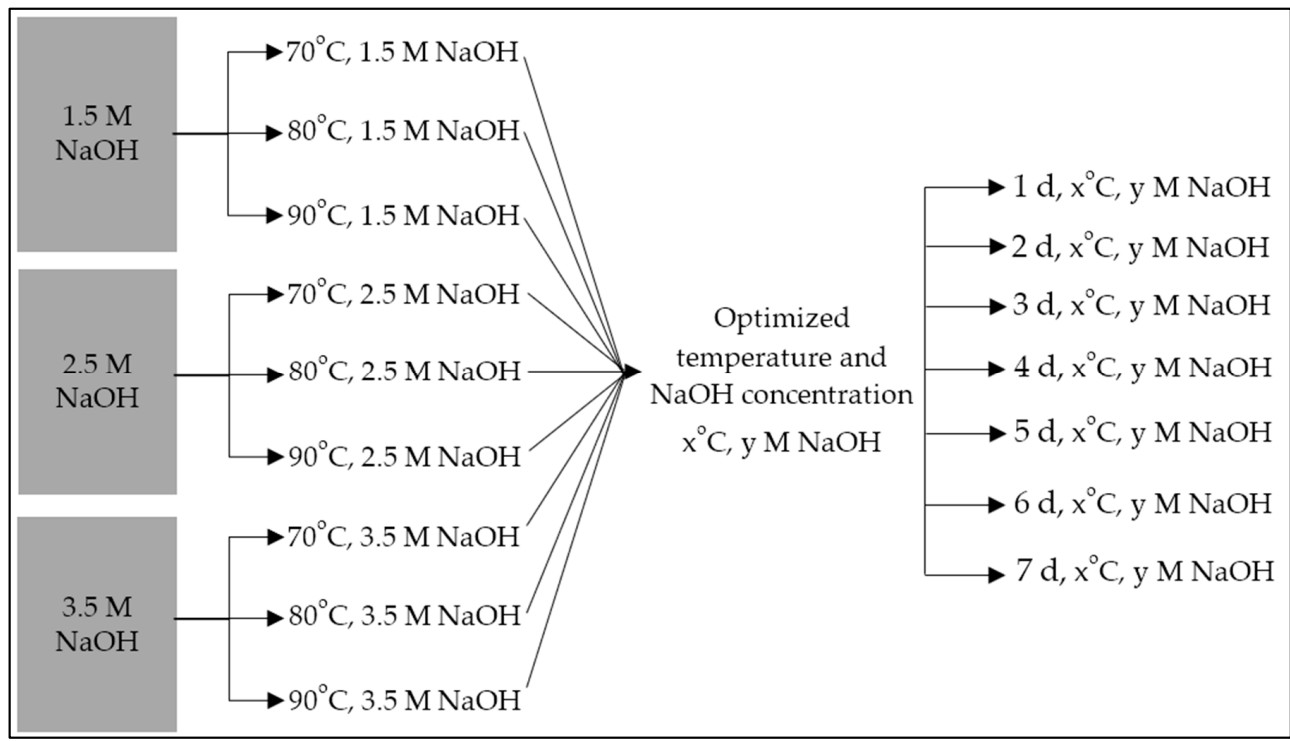

**Figure 1.** Optimization scheme of alkaline hydrothermal treatment of AZLB-Na CHA.

X-ray powder diffraction (XRD) analysis was conducted in a Bruker D8 Advanced instrument (Bruker, Billerica, MA, USA) using Bragg–Brentano Geometry and Copper (Cu) source radiation with 1.54 Å wavelength. The unit crystalline shape of natural AZLB-Na CHA and the treated sample were confirmed by JEOL-JEM-2200FS Energy Filtered Transmission Electron Microscope (JEOL, Tokyo, Japan), featuring an Oxford X-MAX 80 mm$^2$ energy-dispersive X-ray spectrometer (EDS) for element mapping. Fourier transform infrared (FTIR) peak data analysis was conducted using a Cary 630 FTIR, Agilent 600 Series (Agilent, Santa Clara, CA, USA). $^{27}$Al and $^{29}$Si MAS NMR have been recorded on a Bruker AVIII-500 Solid-state NMR spectrophotometer and a Bruker GmbH ePROBE with Biospin software (TopSpin 3.2). The spectral frequencies for $^{27}$Al and $^{29}$Si MAS NMR are 130.3 MHz and 99.3 MHz, respectively.

The concentration of $NH_4^+$ and other relevant cations, e.g., $Na^+$, $Ca^{2+}$, and $Mg^{2+}$, for solutions with low salinity were determined by DIONEX Ion Exchange Chromatography (Model ICS900) (ThermoFisher, Waltham, MA, USA) with an AS40 autosampler installed with it. For determination of $NH_4^+$ concentration in samples with high salinity, a Kjeldahl distillation apparatus was used.

$NH_4^+$ uptake data was obtained through batch experiments where an aliquot of 1.2 g of as-received or modified AZLB-Na CHA was added in a beaker of 1 L of known $NH_4Cl$ solution with constant stirring for 48 h and a contact time long enough to attain equilibrium [29,30]. The pH was maintained at 7.0 by adding drops of a strong acid (HCl) whenever needed. Mass-balance calculations using the initial and final concentration of $NH_4^+$, volume of solution, and mass of zeolite were used to determine the ion-exchange capacity (IEC) of the zeolite. Batch isotherm tests were conducted by varying the zeolite sample dose in aliquots of 0.1–1 g while keeping the $NH_4Cl$ concentration constant. Each sample was stirred for 48 h, providing a contact time long enough to attain equilibrium.

Fixed-bed column studies were conducted in Adjusta Chrome® #11 (Ace Glass Inc., Vineland, NJ, USA) glass column of 1 cm inner diameter and 30 cm length. Five g of dry modified AZLB-Na CHA was made into a slurry using deionized $H_2O$ and poured into the column to attain a bed volume (BV) of 14 mL. A synthetic wastewater influent sample was prepared with the following composition: 89 mg/L sucrose, 165 mg/L NaCl, 25 mg/L $MgSO_4$, 106 mg/L $NH_4Cl$, and 9 mg/L $Na_2HPO_4$. The corresponding cation concentration was 2.93 meq/L of $Na^+$, 1.55 meq/L of $NH_4^+$, and 0.52 meq/L of $Mg^{2+}$. The synthetic wastewater was injected into the column using a synchronous FMI lab pump, Model QSY (Fluid Metering Inc., Syosset, NY, USA). The pump flow rate was maintained at 4.5 mL/min and the empty bed contact time (EBCT) of the column was 3 min. Column effluent was collected by a Spectra/Chrome® CF-1 Fraction collector, Wazobia Enterprise, Houston, TX, USA (Spectrum Chromatography). Column breakthrough BV was identified when column effluent $NH_4^+$ concentration reached 10% of its influent value. The exhausted column was regenerated with 2.0 M NaCl at the same flow rate and EBCT as during the exhaustion run.

## 3. Results and Discussion

In Figure 1, two steps of optimization have been illustrated; (i) nine samples were prepared with varying NaOH concentrations and solution temperatures, and (ii) identifying the suitable NaOH concentration and temperature, seven more samples were prepared with a varying extent of hydrothermal treatment, keeping the other two prior-concluded parameters consistent. In Table 1, the chemical composition of all those first nine treated AZLB-Na CHA samples has been presented. This chemical-composition data has been collected from six different points of each sample.

The Si/Al ratio is a key feature of zeolite. A lower Si/Al ratio ensures more accommodation for an exchangeable extra-framework cation. This results in a higher cation exchange capacity [31–34]. During alkaline hydrothermal treatment, zeolite undergoes selective desilication of the framework. This further supports the reinsertion of extra-framework Al (EFAL) into the framework [23,35,36]. However, this interesting phenomenon has several scopes of rigorous research. Thus, zeolite with a low Si/Al ratio is a favorable choice for

an ion-exchange mechanism. In this work, the target was to reach a Si/Al ratio < 2 in treated AZLB-Na CHA, because this zeolite with Si/Al < 2 is known as 'low Si/Al ratio zeolite' which possesses greater CEC. From Table 1, it is evident that AHTCHA$^{5D70C2.5M}$ transitioned into a low silica (Si/Al $\leq$ 2) zeolite from an intermediate silica (2 $\leq$ Si/Al $\leq$ 5) zeolite [37]. The EDS data for AZLB-Na CHA and AHTCHA$^{5D70C2.5M}$ presented in Table 1 suggest that desilication occurred in the parent zeolite during alkaline hydrothermal treatment. AZLB-Na CHA, initially an intermediate Si zeolite (2 < Si/Al < 5) with Si/Al = 3.2, transformed into a low-silica zeolite (Si/Al $\leq$ 2) [37]. AHTCHA$^{5D70C2.5M}$, AHTCHA$^{5D70C3.5M}$, AHTCHA$^{5D80C2.5M}$, AHTCHA$^{5D80C3.5M}$, AHTCHA$^{5D90C2.5M}$, and AHTCHA$^{5D90C3.5M}$ have been selected as per this condition. Because AZLB-Na CHA is a Na-loaded natural CHA, i.e. its exchangeable cation is Na, and maximizing Na$^+$ loading through alkaline hydrothermal treatment was another goal. Those previously selected six samples can be further arranged in the following order: AHTCHA$^{5D80C3.5M}$ > AHTCHA$^{5D70C2.5M}$ > AHTCHA$^{5D70C3.5M}$ > AHTCHA$^{5D90C2.5M}$ $\approx$ AHTCHA$^{5D90C3.5M}$ > AHTCHA$^{5D80C2.5M}$. However, a significant amount of AZLB-Na CHA was destroyed into slurry form due to the corrosive nature of the 3.5 M NaOH solution at a higher temperature. Considering the feasibility, the second highest Na$^+$ loaded sample, AHTCHA$^{5D70C2.5M}$, was chosen as the most desirable sample among six previously selected samples.

**Table 1.** Chemical composition of nine samples at the first step of treatment optimization.

| Element \ Sample | AZLB-Na CHA | AHTCHA$^{1D70C1.5M}$ | AHTCHA$^{1D70C2.5M}$ | AHTCHA$^{1D70C3.5M}$ | AHTCHA$^{1D80C1.5M}$ | AHTCHA$^{1D80C2.5M}$ | AHTCHA$^{1D80C3.5M}$ | AHTCHA$^{1D90C1.5M}$ | AHTCHA$^{1D90C2.5M}$ | AHTCHA$^{1D90C3.5M}$ |
|---|---|---|---|---|---|---|---|---|---|---|
| O | 60.41 | 67.53 | 66.33 | 67.15 | 67.40 | 66.73 | 68.81 | 67.79 | 69.42 | 66.51 |
| Na | 6.31 | 6.22 | 9.25 | 9.08 | 6.93 | 7.278 | 9.55 | 6.66 | 9.026 | 9.02 |
| Si | 25.35 | 18.31 | 13.98 | 13.78 | 17.59 | 14.62 | 12.77 | 17.44 | 11.44 | 13.42 |
| Al | 7.91 | 7.95 | 10.45 | 9.98 | 8.08 | 11.28 | 8.86 | 8.10 | 10.12 | 11.05 |
| Si/Al | 3.2 | 2.303 | 1.34 | 1.38 | 2.18 | 1.30 | 1.44 | 2.15 | 1.13 | 1.21 |

Figure 2 illustrates the FTIR spectra of modified chabazites treated in a 2.5M NaOH solution at 70 °C for 1, 2, 3, 4, 5, 6, and 7 days. Previous studies have assigned specific vibrational modes to the infrared spectrum. For instance, Si to Si-O asymmetric stretches are typically identified in the 950–1250 cm$^{-1}$ range, symmetric Si-Si stretching in the 600–800 cm$^{-1}$ range, and Si-O bending and distortion modes below 600 cm$^{-1}$ [38–41]. Si to Si-O asymmetric stretch is clearly visible in the presented FTIR. The symmetric Si-Si stretching, Si-O bending and distortion mode are illustrated in the inset of Figure 2. Moreover, the distinctive FTIR peak characterizing the double six-membered ring (D6R) of CHA typically appears in the range of 570–635 cm$^{-1}$ [38,40,42]. For AZLB-Na CHA, two clear peaks were evident at 522 cm$^{-1}$ and 640 cm$^{-1}$, as shown in Figure 2. However, these peaks were absent in all treated samples. Thus, it can be inferred that alkaline hydrothermal treatment plays a significant role in framework destruction, potentially causing alterations to the zeolite phase.

Additionally, OH groups, which may be affected by the low water-vapor pressure, are observed at a wavelength of approximately 2460 cm$^{-1}$ [39]. This peak is observed in hydrothermally treated AZLB-Na CHA samples at approximately 2360 cm$^{-1}$.

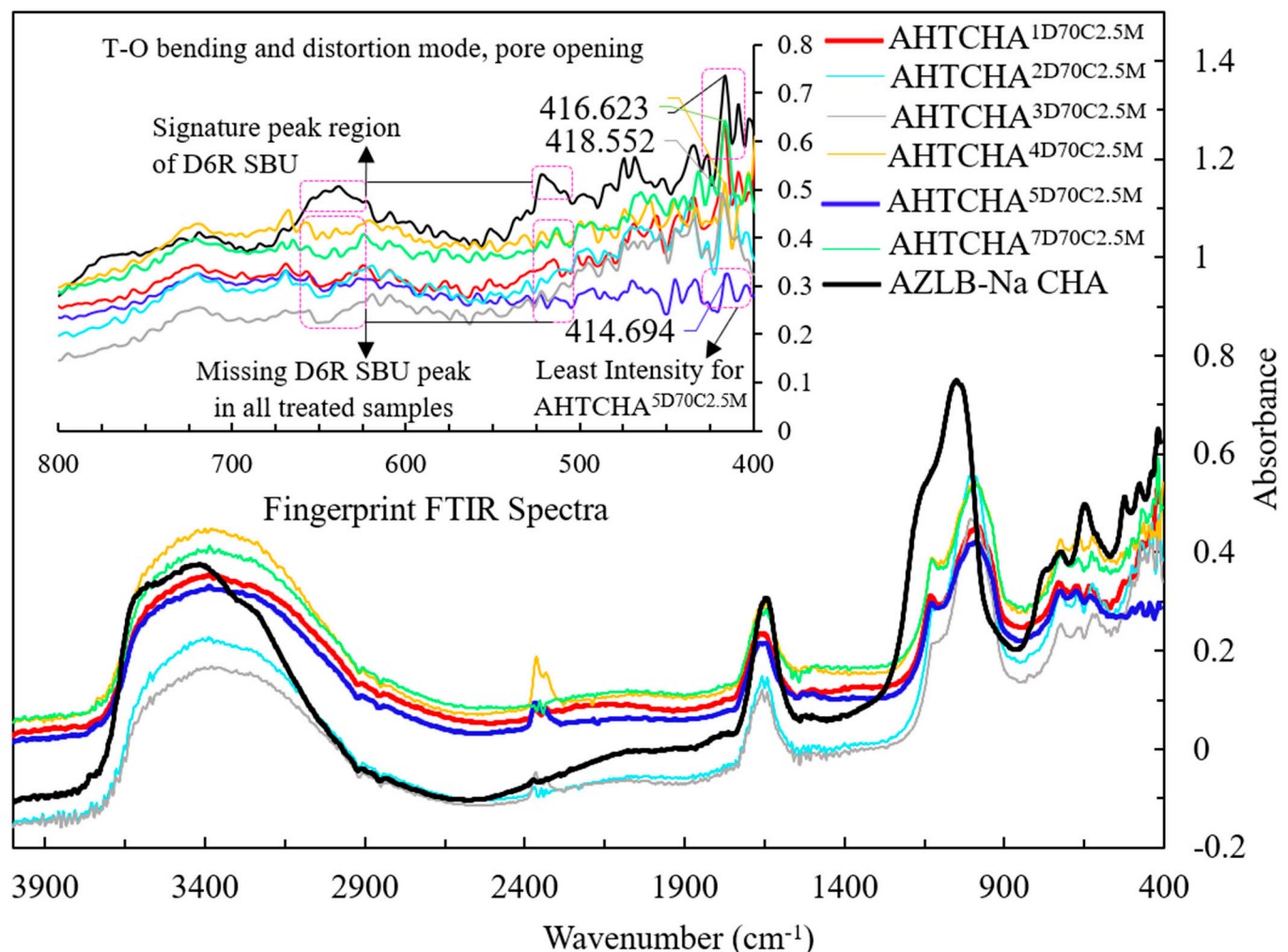

**Figure 2.** FTIR spectra of seven treated samples and AZLB-Na CHA.

For zeolite, the FTIR active band for adsorbed $H_2O$ is typically in the 1640–1650 $cm^{-1}$ range [39]. This band broadens with increasing water-vapor pressure [39]. Extended alkaline hydrothermal treatment (3–7 d) resulted in increased water-vapor pressure on the AZLB-Na CHA, causing this band to widen.

In the range of 3300–3600 $cm^{-1}$, an intense peak for O-H stretching of the zeolite was evident [43]. For the AZLB-Na CHA, the peak at 3568 $cm^{-1}$ was attributed to the OH vibration of eight-membered rings [39,41].

The peak in the spectral range of 300–420 $cm^{-1}$ is linked to the pore-opening characteristic of the zeolite, playing a crucial role in the framework-structure sensitivity [38]. In Figure 2, a distinct, sharp peak at 414 $cm^{-1}$ is evident for the AZLB-Na CHA sample. However, for the treated samples, this peak exhibited a lower intensity. At this wavelength, AHTCHA$^{5D70C2.5M}$ exhibited the lowest intensity, leading to speculation that, during alkaline hydrothermal treatment, the destruction of the framework potentially leads to the formation of a denser channel network characterized by a narrower pore opening. This may be the reason for the observed changes in intensity in this spectral range.

The XRD analysis of AZLB-Na CHA is presented in Figure 3a. The diffraction pattern is represented over an angle range of 2θ = 0–50° on the x-axis; the y-axis displays the relative intensity, considering the highest peak count as 100.

These peak patterns were comparable with the International Zeolite Association (IZA)-reported peaks of natural CHA [6]. However, an offset of 2θ = 1.1–4.5° has been observed compared to IZA-reported peaks. This can be attributed to differences in the

XRD instruments used, variations in the instrumentation parameters, potential specimen displacement errors, and the degree of crystallinity of the two samples.

Figure 3a–c shows the XRD scan of AZLB-Na CHA and two modified AZLB-Na CHA samples. These treated specimens underwent alkaline hydrothermal treatment of AZLB-Na CHA for durations of 1 d and 5 d, respectively, maintaining the NaOH solution concentration of 2.5 M and a temperature of 70 °C in both cases. Comparison between the XRD of AZLB-Na CHA, well-indexed with JCPDS card 34-137, in Figure 3a and treated samples in Figure 3b,c, shows that certain peaks in the alkali-mediated hydrothermally treated samples are considerably sharper than those in the AZLB-Na CHA. Those peaks have been labeled in Figure 3a. In XRD, a fingerprint peak at $2\theta = 4.95°$ is commonly observed in zeolites with $Na^+$ as the exchangeable cation. The parent sample used in this study (AZLB-Na CHA) primarily contained $Na^+$ as the exchangeable cation, reflected in the peak at $2\theta = 4.95°$. There were no significant changes in the position and intensity of the major peaks in AZLB-Na CHA during alkaline treatment. However, the doublet at $2\theta = 4.95°$ exhibited a considerable increase in intensity. The intensity of the diffraction peaks was correlated with the plane indices, indicating the angle in the diffraction patterns. Long sharper peaks often indicate lower plane indices, suggesting narrower pattern angles. A potential explanation for these sharper peaks is the second-order transformation of AZLB-Na CHA during alkaline treatment. Second-order transformation is a displacive phenomenon involving the change of one crystal system to another, such as from trigonal-hexagonal [44] to cubic [45] or triclinic.

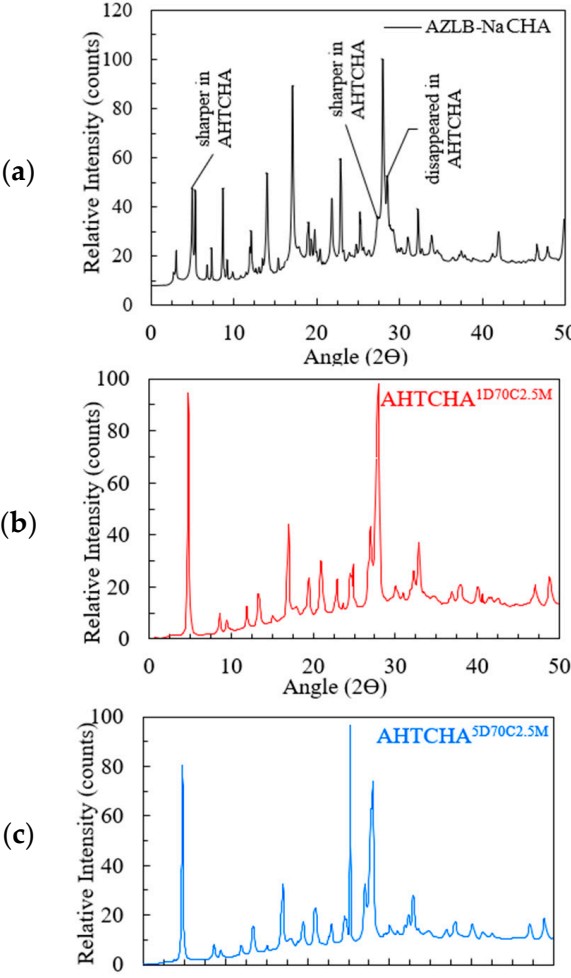

**Figure 3.** (**a**) XRD of AZLB-Na CHA; (**b**) XRD of AHTCHA$^{1D70C2.5M}$; (**c**) XRD of AHTCHA$^{5D70C2.5M}$.

Scherrer [46] proposed a correlation between the peak broadening and mean crystallite size using the following expression:

$$B = K\lambda/(L\,cos\theta) \tag{1}$$

where *B* = broadening of the peak in radians, *L* = mean crystallite size, *K* = shape factor, usually 0.9, *λ* = wavelength, and *θ* = angle between the incident ray and the scattering plane in degrees.

According to Equation (1), there is an inverse relationship between the peak broadening and mean crystal size [46]. The peak at 2θ = 27.96° for AHTCHA appears broader than that for AZLB-Na CHA at the same position, indicating a reduction in crystal size due to alkaline hydrothermal treatment. In the XRD patterns of the alkaline-mediated hydrothermally treated AZLB-Na CHAs shown in Figure 3b,c, the peaks at 2θ = 3.05° and 35.99° are absent. As the particle size decreased, the peaks broadened significantly. In some cases, the peaks may be too broad to be distinctly discerned. The absence of peaks at 2θ = 3.05° and 35.99° in the XRD of AHTCHA$^{1D70C2.5M}$ and AHTCHA$^{5D70C2.5M}$ suggests a reduction in crystal size during alkaline hydrothermal treatment.

A sharp, newly formed peak at 2θ = 25.15° is observed in Figure 3c, a characteristic peak of analcime (ANA) zeolite. This peak was faint in AHTCHA$^{1D70C2.5M}$ at 2θ = 24.88° but was the most prominent peak in AHTCHA$^{5D70C2.5M}$, possibly due to longer exposure of AZLB-Na CHA in the experimental environment during preparation. Additionally, a few insignificant peaks of AZLB-Na CHA disappeared in Figure 3b,c; the plot appears much smoother for AHTCHA$^{5D70C2.5M}$ than for AHTCHA$^{1D70C2.5M}$. The pattern indicates alterations in the crystal structure and composition due to the longer treatment duration, resulting in changes in the XRD pattern and the emergence of a distinct peak associated with a different zeolite phase.

A powder-pattern identification table [47–49] was used to comprehend the AZLB-Na CHA framework. Traditionally, the crystal structure of ANA has been classified as a cubic space group. However, studies [50,51] have reported deviations from cubic systems in many ANA structures. Revisiting the XRD results to identify the crystal structure of AHTCHA$^{5D70C2.5M}$, the sharpest peak at 2θ = 25.44° in Figure 3b matches the characteristic peak of analcime (ANA). Additionally, two significant peaks at 2θ = 17.01° and 27.96° also indicate ANA, with relative intensities of 34% and 75.4%, respectively.

From data recorded by the International Zeolite Association (IZA) for ANA, the sharpest peak is at 2θ = 25.936°; other significant peaks are at 2θ = 15.797° and 30.513°, with relative intensities of 60.75% and 51.4%, respectively. The American Mineralogist Crystal Structure Database compiled by Bob Downs and Paul Hesse at the University of Arizona contains multiple ANA diffraction patterns [52]. The XRD results obtained in this study align with ANA having a P-1 crystal-space group, with the sharpest peak at 2θ = 26.4° and significant peaks at 2θ = 16.06° and 30.89°, as reported by multiple scholars [53,54]. The XRD peaks show a maximum shift of Δ2θ = 2.6° and 2.93° from the calculated pattern published in IZA and the American Mineralogist Crystal Structure Database record for ANA, respectively. The space group P-1 belongs to the centrosymmetric triclinic crystal structure with three unequal axis lengths and planar angles, further confirming the primary conclusion regarding the AHTCHA$^{5D70C2.5M}$ crystal system. The crystal structure of AHTCHA$^{5D70C2.5M}$ can be well-indexed into a Joint Committee of Powder Diffraction Standards (JCPDS) JCPDS 41-1478 XRD card.

The transmission electron microscopy (TEM) image in Figure 4a aligns with the conclusion regarding the crystal lattice system of AZLB-Na CHA. The hexagonal faces of the crystal are shown in Figure 4b. The TEM image of the AZLB-Na CHA crystal in Figure 4c indicates that one dimension is longer than the others. Although it is challenging to determine the planar angles from a TEM image (TEM reflects the projection of a crystal), it is inferred from the hexagonal projection and unequal axial dimensions shown in Figure 4a–c that the AZLB-Na CHA crystal is part of a trigonal–hexagonal crystal system.

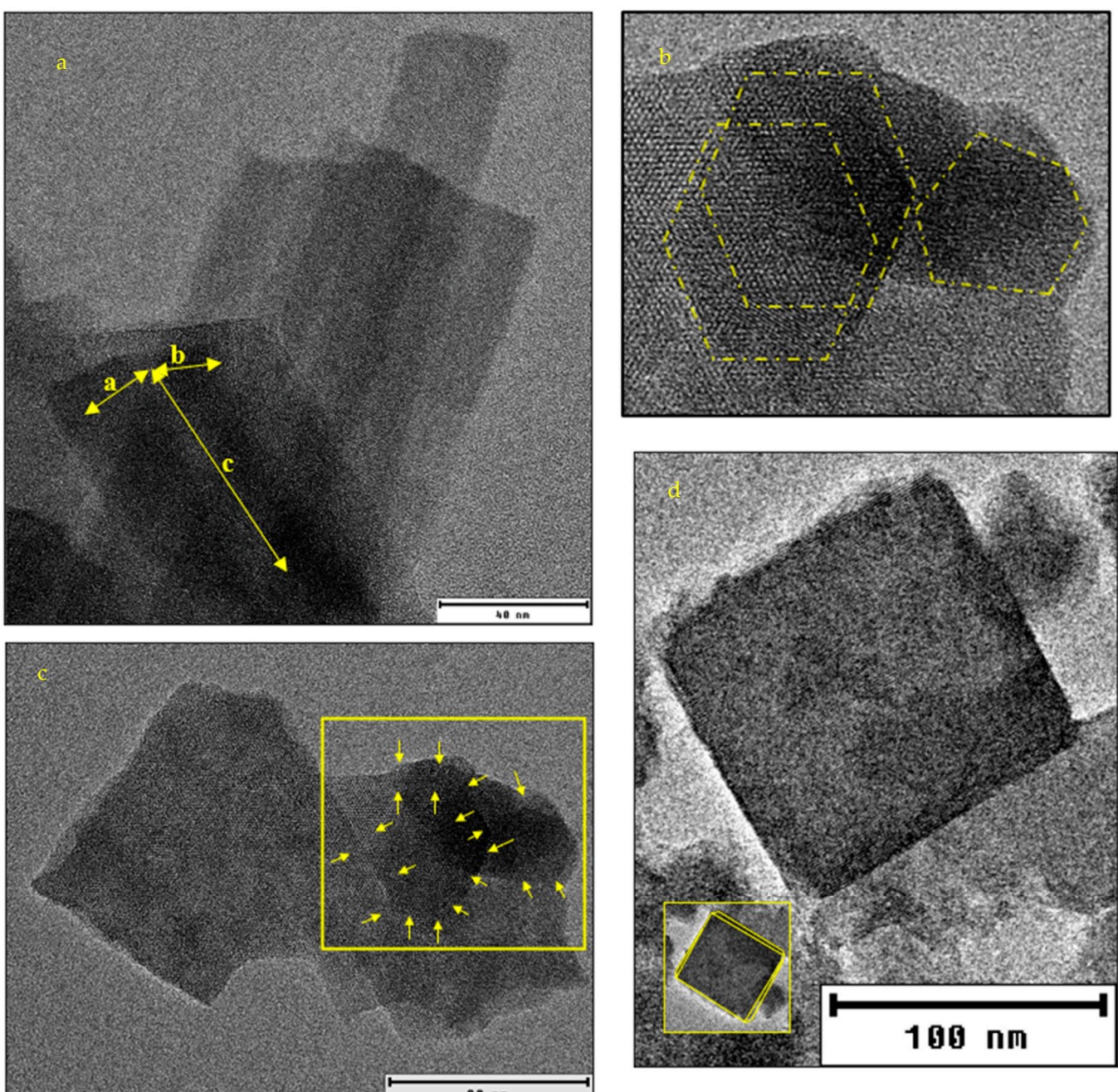

**Figure 4.** (**a**) HRTEM image of AZLB-Na CHA with visible edges marked with yellow highlighter. (**b**) zoomed section of (**a**) with highlighted hexagonal projection of AZLB-Na CHA crystal. (**c**) unequal dimension of AZLB-Na CHA crystal. (**d**) HRTEM image of AHTCHA$^{5d70C2.5M}$.

Figure 4d shows a TEM image of AHTCHA$^{5D70C2.5M}$. The grain boundaries yellow highlighted in the crystal may represent a 2D projection of a triclinic crystal. In Figure 4d, the yellow highlighted edge of the AHTCHA$^{5D70C2.5M}$ crystal exhibits a slight deviation from 90° in its planar angle. This deviation shows a fundamental characteristic of a triclinic crystal system, where $\alpha \neq \beta \neq \gamma \neq 90^o$. In addition, the two axial lengths in this image are slightly different. Even a small discrepancy in axial lengths (one-thousandth of an Å) aligns with the defining features of a triclinic crystal system where $a \neq b \neq c$ [52].

Figure 5 shows the $^{27}$Al MAS NMR scan of the AZLB-Na CHA and three other modified samples where the concentration of the alkali is different, i.e., the time of treatment and the temperature are the same. A low-silica zeolite generally contains an aluminum-

dense framework, indicated by the Si magic-angle spinning nuclear magnetic resonance (MAS NMR) peaks shown in Figure 6 for the four different samples. The Si and Al MAS NMR data provided significant insights into the zeolite structure.

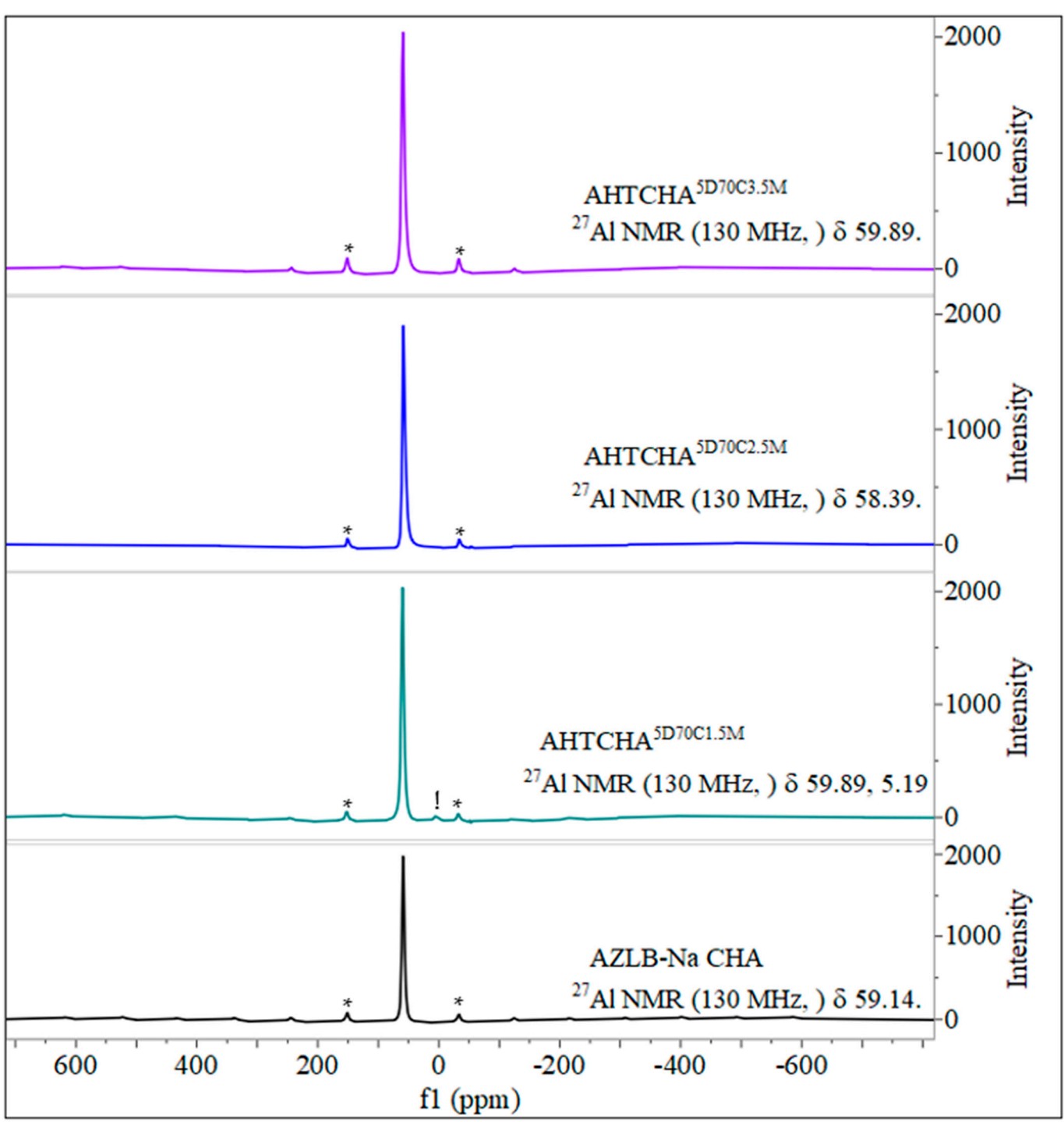

**Figure 5.** Solid state $^{27}$Al MAS NMR of AZLB-Na CHA and AHTCHA$^{5D70C1.5M}$, AHTCHA$^{5D70C2.5M}$, and AHTCHA$^{5D70C3.5M}$.

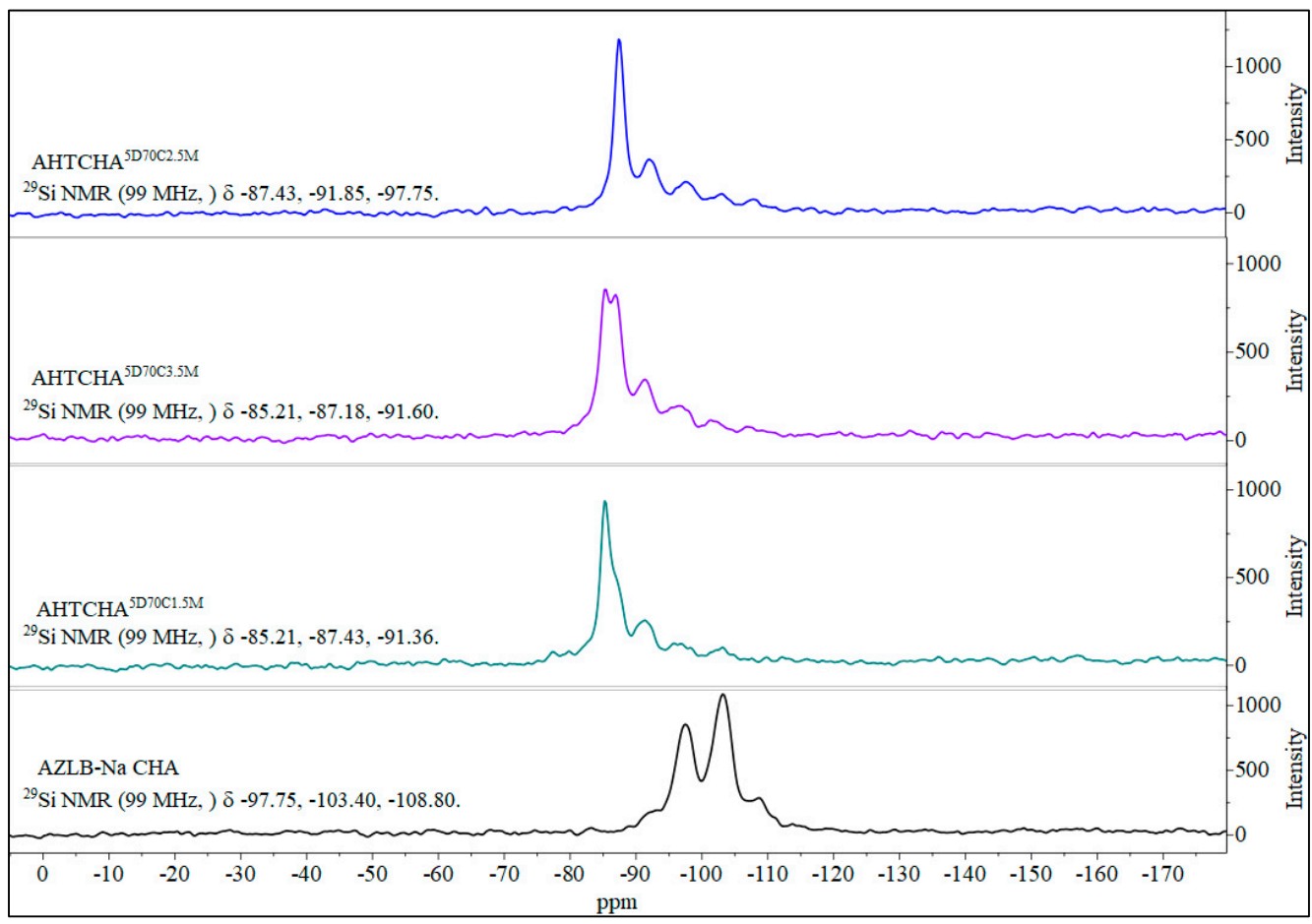

**Figure 6.** Solid-state $^{29}$Si MAS NMR of AZLB-Na CHA and AHTCHA$^{5D70C1.5M}$, AHTCHA$^{5D70C2.5M}$, and AHTCHA$^{5D70C3.5M}$.

In Al MAS NMR, the tetrahedrally coordinated framework of Al resonates in the range of 50–65 ppm, whereas octahedrally coordinated extra-framework Al typically appears as a resonance line at approximately 0 ppm [55]. As illustrated in Figure 6, the solid-state $^{27}$Al MAS NMR resonance peak in the AHTCHA$^{5D70C1.5M}$ sample was observed at approximately 0 ppm (5.19 ppm, marked with an exclamation point) alongside spinning sidebands (marked with asterisks). This observation indicates that treatment. However, in the other two samples, AHTCHA$^{5D70C2.5M}$ and AHTCHA$^{5D70C3.5M}$, apart from the pair of spinning sidebands, no distinct signal was observed at approximately zero ppm. This absence may be explained by the reinsertion of the extra-framework Al into the lattice, potentially leading to the reformation of an existing zeolitic crystal framework into a different structure. Realumination during hydrothermal treatment of zeolites has been observed [56,57].

Figure 6 shows the solid-state $^{29}$Si MAS NMR spectra of the four samples, demonstrating the distinction between the AZLB-Na CHA and the three alkaline hydrothermally treated samples. This difference suggests a significant alteration in the framework during the alkaline hydrothermal treatment of AZLB-Na CHA. Fyfe et al. [58] conducted extensive studies on the solid-state $^{29}$Si and $^{27}$Al MAS NMR spectra of zeolites.

They identified five possible local environments of a silicon atom: Si(OAl)$_4$, Si(OAl)$_3$(OSi), Si(OAl)$_2$(OSi)$_2$, Si(OAl)(OSi)$_3$, and Si(OSi)$_4$. The corresponding chemical-shift ranges of these orientations are −82.5−−87.5, −88−−95, −93−−100, −97−−105 and −103−−115 ppm, respectively [58]. In the AZLB-Na CHA, Si primarily occurred in the Si(1Al), Si(2Al), and Si(3Al) orientations; Si(2Al) was the most frequent. However, after alkaline hydrothermal treatment, the AZLB-Na CHA transitioned into a Si(4Al) form, represent-

ing the most aluminum-dense form of the zeolite crystal. Of the three treated samples, AHTCHA$^{5D70C2.5M}$ exhibited the highest peak intensity corresponding to Si(4Al), indicating that 5-d alkaline hydrothermal treatment of AZLB-Na CHA in 2.5 M NaOH solution at 70 °C was the optimal condition to reverse the dealumination process initiated by this treatment. Cheetham et al. [59] extensively analyzed MAS NMR and XRD data to tentatively locate Na$^+$ ions in Na-loaded CHA. They marked those positions as SI—at the center of the hexagonal prism, SII—at the 6-membered ring window of the prism, SIII—at the 8 membered ring window close to corner of the 4 membered ring and SIII′—almost at the center of the 8 membered ring. marking these positions as SI, SII, SIII, and SIII′. In Na-loaded CHA, Na$^+$ is preferentially situated at the SII, SIII, and SIII′ locations. As further Na$^+$ loading occurs in the AZLB-Na CHA during alkaline hydrothermal treatment, the collapse of the double six-membered ring poses no challenge to Na$^+$ location in the reoriented zeolitic crystal. The disappearance of the characteristic peaks in the FTIR data in Figure 2, specific to the secondary building unit (SBU) double six-membered ring or hexagonal prism, in all AHTCHA samples signified the second-order transformation of AZLB-Na CHA into ANA during treatment. ANA lacked the D6R SBU characteristics of CHA, resulting in the absence of these peaks [45].

*Ion-Exchange Isotherm*

AHTCHA$^{5D70C2.5M}$ was used for all subsequent experiments. From ion-exchange isotherm experiments, its IEC was calculated from mass balance to be 4.19 meq/g. During ion exchange, NH$_4^+$ ion replaced the extra-framework Na$^+$ ion that eluted into the experimental solution. When contrasted with the IEC of the as-received chabazite (AZLB-Na CHA) of 2.12 meq/g, it is clear that the alkaline hydrothermal treatment protocol employed almost doubled the IEC. The significant enhancement of the IEC makes it a promising candidate for NH$_4^+$ removal from wastewater. The results of the batch isotherm experiments of NH$_4^+$ uptake with AHTCHA$^{5D70C2.5M}$ were fitted to the Freundlich, Langmuir-1, Henry, Temkin in Figure 7a–d, and modified Langmuir isotherm [60,61] in Figure 8. According to the modified Langmuir Isotherm (Figure 8)

$$q_e = \frac{q_m K_{ML} C_e}{(C_s - C_e) + K_{ML} C_e} \tag{2}$$

where

$q_e$ = equilibrium uptake (meq/g)
$q_m$ = maximum uptake (meq/g)
$K_{ML}$ = Modified Langmuir Equilibrium constant (dimensionless)
$C_e$ = equilibrium concentration (meq/L)
$C_s$ = solution concentration when the solution is saturated (meq/L)

This equation is linearized thus:

$$\frac{C_e}{q_e} = \frac{(K_{ML} - 1)}{q_m K_{ML}} C_e + \frac{C_s}{q_m K_{ML}} \tag{3}$$

According to this model, the uptake of solute depends on two important parameters: (i) the bulk adsorbate concentration in the solution and (ii) the availability of vacant sites on the sample.

The breakthrough curve of the fixed bed containing the modified chabazite that was fed a synthetic wastewater sample is shown in Figure 9. The composition of synthetic wastewater is provided in Table 2. The break point (BP) and exhaustion point (EP) were chosen to be when the effluent NH$_4^+$ concentration was 10% and 90% of the influent NH$_4^+$ concentration, respectively. When the AHTCHA$^{5D70C2.5M}$ fixed bed was exhausted, it was regenerated with a 2.0 M NaCl solution. The regeneration curve is shown in Figure 10. It may be noted from the sharpness of the effluent profile that regeneration is highly efficient, with the recovery of >94% of the NH$_4^+$ (loaded on the exhausted AHTCHA$^{5D70C2.5M}$)

in <10 BV. From the type of breakthrough curve, especially due to the gradual rise, it is evident that it has excellent CEC, and it gets exhausted gradually. A similar kind of performance of ion-exchange materials in a packed-bed column has been outlined by several researchers [62–64]. The regeneration profile (Figure 10) provides solid evidence of a concentrated ammonium solution that could be used as a stand-alone nitrogenous fertilizer or combined with potassium and phosphorus to create a commercial fertilizer. Admittedly, ammonium has to be separated from the saline regenerant solution (e.g., by distillation and recondensation) in order for it to be used as a fertilizer, but this is outside of the scope of this article. The main purpose of this article is to provide evidence that (1) the modified chabazite can be used as a selective ammonium ion-exchanger to remove ammonium from a solution that contains competing cations and (2) regeneration of this zeolite provides an ammonium-rich solution. The zeolite can be reused in the next exhaustion cycle. Studies conducted for three cycles (each cycle consists of an exhaustion and a regeneration step) have shown minimal loss of modified chabazite's ion-exchange capacity, ammonium selectivity, or regeneration efficiency.

**Table 2.** Synthetic wastewater characteristics.

| Chemical | Conc. (mg/L) | Cation | Conc. (meq/L) |
|---|---|---|---|
| NaCl | 167.2 | Na$^+$ (from NaCl and Na$_2$HPO$_4$) | 2.93 |
| Na$_2$HPO$_4$ | 10.2 | | |
| MgSO$_4$ | 26.1 | Mg$^{2+}$ | 0.53 |
| NH$_4$Cl | 107.0 | NH$_4^+$ | 1.55 |
| C$_{12}$H$_{22}$O$_{11}$ Sucrose | 89.4 | | |
| pH | 7.4 | | |

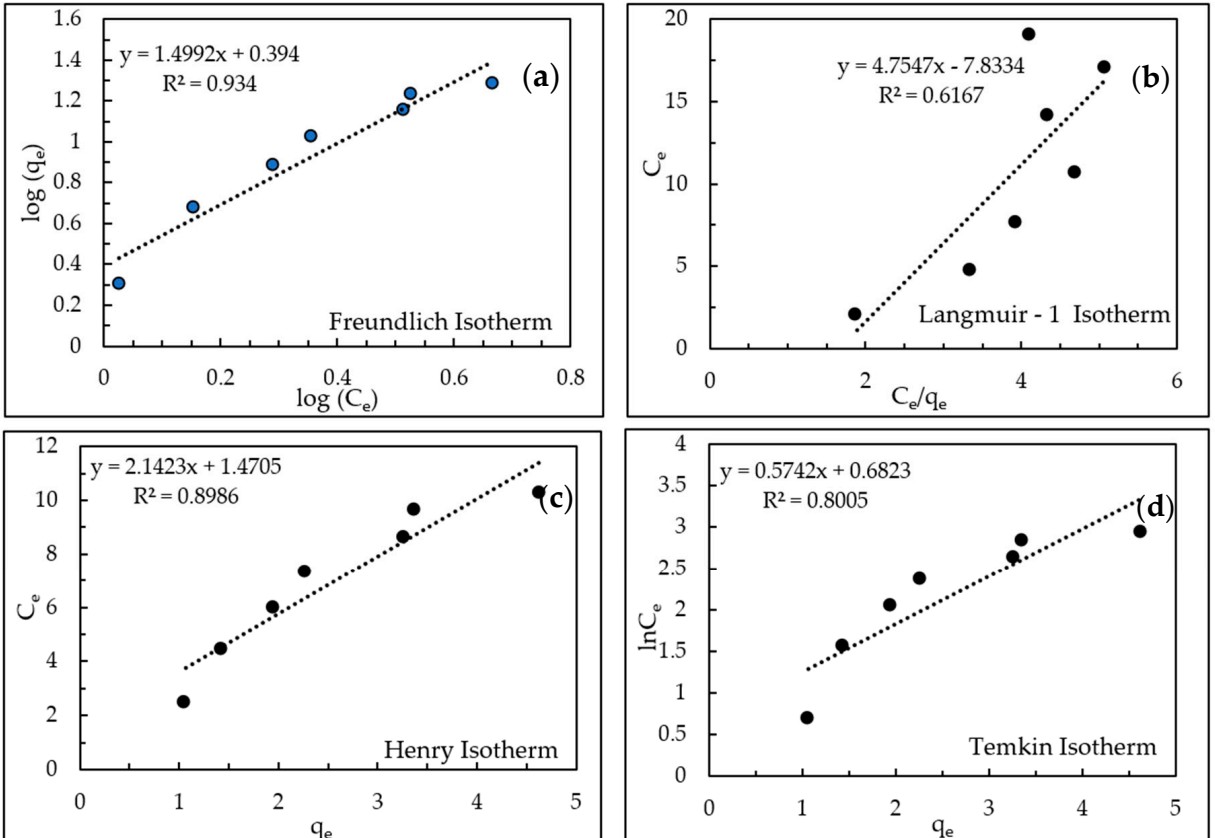

**Figure 7.** (**a**) Freundlich isotherm, (**b**) Langmuir-1 isotherm, (**c**) Henry isotherm, (**d**) Temkin isotherm.

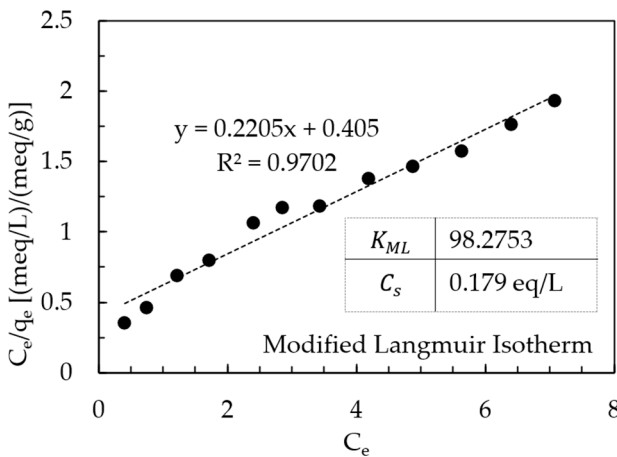

**Figure 8.** Modified Langmuir isotherm plot.

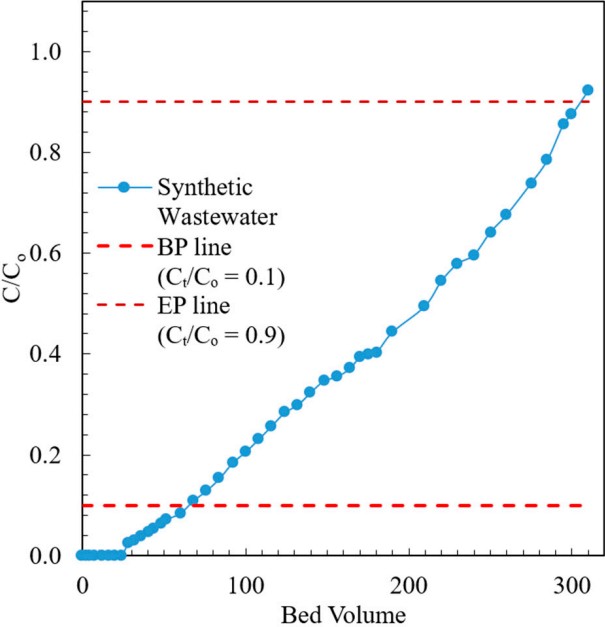

**Figure 9.** Effluent profile of modified chabazite fixed-bed column fed with synthetic wastewater.

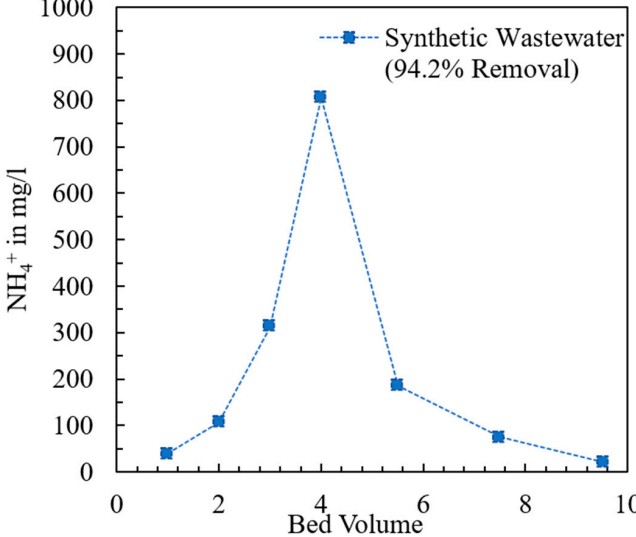

**Figure 10.** Regeneration of exhausted fixed-bed column containing modified chabazite.

## 4. Conclusions

Alkaline hydrothermal treatment of AZLB-Na CHA was conducted to enhance its cation exchange capacity (CEC). The resulting modified AZLB-Na CHA, AHTCHA$^{5D70C2.5M}$, was used for selective ion exchange, specifically to remove $NH_4^+$ from wastewater. Optimized experimental parameters, including the concentration, temperature of the alkaline solution, and duration of exposure to AZLB-Na CHA, were established to produce the most effective modified sample in terms of maximizing the CEC.

Environmental researchers are exploring similar methods to enhance the properties of zeolites, with comparable objectives. However, comprehensive studies analyzing the physical and chemical characteristics of modified zeolites after alkaline hydrothermal treatment are limited; the improved characteristics of the modified zeolite are often attributed to the crystal-space group of the parent zeolite. In fact, the enhanced physicochemical behavior resulting from alkaline hydrothermal treatment of zeolites is often specific to the crystal-space group of the resulting zeolite. This study underscores the necessity of thorough physicochemical analyses of zeolites after alkaline hydrothermal treatment.

This research investigated the effects of alkaline hydrothermal treatment on a CHA crystal lattice. Analytical data established that such a treatment can induce a displacive transformation in CHA. This study comprehensively determined the final crystal structure of the modified zeolite and other significant alterations caused by the treatment. As a future aspect of this work, observation should be made about the zeolitic framework changes and performance changes due to this, followed by the successive $NH_4^+$ removal.

The modified chabazite (after alkaline hydrothermal treatment) can be used in a fixed-bed column configuration to selectively remove ammonium from a solution that contains competing cations. Regeneration of the exhausted zeolite provides an ammonium-rich solution that can be used as a stand-alone fertilizer or as a component of a total fertilizer. The zeolite can be reused in the next exhaustion cycle. Studies conducted for multiple cycles (each cycle consists of an exhaustion and a regeneration step) have shown no loss of modified chabazite's ion-exchange capacity, ammonium selectivity, or regeneration efficiency.

**Author Contributions:** Conceptualization, S.S.; Methodology, S.S.; Validation, D.D.; Formal Analysis, D.D.; Investigation, D.D.; Resources, S.S.; Data Curation, D.D.; Writing—Original Draft Preparation, D.D.; Writing—Review and Editing, D.D.; Visualization, S.S.; Supervision, S.S.; Project Administration, S.S.; Funding Acquisition, S.S. All authors have read and agreed to the published version of the manuscript.

**Funding:** Partial support for this project was provided by the NSF (Grant CBET 1511399) and the US-Israel Binational Agricultural Research and Development Fund (Grant US-5439-21).

**Data Availability Statement:** The data that support the findings of this study are available from the corresponding author, S.S., upon reasonable request.

**Acknowledgments:** We extend our gratitude to St. Cloud Mining for generously providing the chabazite samples for this research. Special thanks to James Golen of the University of Massachusetts, Dartmouth, and Milan Gembicky and Han Ngyuen from the University of California, San Diego, for their invaluable contributions to the XPD analysis of our samples. We are also deeply appreciative of Alexander Ribbe from the University of Massachusetts, Amherst, for conducting high-resolution transmission electron microscopy (HRTEM), and Caitlin Quinn from the University of Delaware for conducting solid-state magic-angle spinning nuclear magnetic resonance (MAS NMR) analysis of our samples.

**Conflicts of Interest:** The authors declare no conflicts of interest.

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
