# Peer review of "Alkaline Hydrothermal Treatment of Chabazite to Enhance Its Ammonium Removal and Recovery Capabilities through Recrystallization"

_processes, doi:10.3390/pr12010085_

Round 1

Reviewer 1 Report

Comments and Suggestions for Authors

To Sustainability

Title: Alkaline Hydrothermal Treatment of Chabazite to Enhance Its Ammonium Removal and Recovery Capabilities

Ref.: Processes-2757132

Dear Editor

Das and Sengupta report research in which they treated zeolite by alkaline hydrothermal treatment to improve ammonium removal and recovery capacity. The author used chabazite, a natural zeolite, for alkaline hydrothermal treatment at different alkali concentrations, temperatures and times. The author studied XPD, FTIR, TEM, MAS-NMR and analytical techniques to identify the chabazite before and after treatment and reported the transformation of analcime after hydrothermal treatment. The obtained modified zeolite is used for the removal and recovery of ammonium from synthetic wastewater. Furthermore, the regeneration of the zeolite produced an NH4+-rich solution that can be used as a fertilizer. I see that the author managed to characterize and study the behavior in detail. What they have in common is that the transition works smoothly and well. However, the authors need to add a paragraph in the introductory section that illustrates the novelty of the current experiments and provides a clearer goal that the reader can easily follow. I believe that the work described is sufficient for this journal. Therefore, it is expected that this article will be accepted with minor comments.

The detailed Minor comments are listed as following:

1. Introduction- need to add paragraph with state of art novelty in this paper

2. . Is it possible for the author to provide BET surface area and pore size since the author claimed that the framework changed after treatment and therefore the surface phenomenon used for adsorption needs to be confirmed??

3. Figure 2 is very informative, but the author did not describe it in detail. It is recommended to present the numbers obtained, as well as their meaning and variations/shifting of peaks, in more detail.

4. Synthetic wastewater experiments data need error bar and also need to report the mean value of the experiments. 

Reviewer 2 Report

Comments and Suggestions for Authors

The authors presented that alkaline hydrothermal treatment of chabazite caused a significant alteration in the crystal structure and analyzed through a series of tests. This method enhanced the cation exchange capacity of chabazite, ammonium removal and recovery capabilities. However, there is insufficient research on adsorption. There are still some issues that need to be answered and supplemented.

1.Is there any TEM images from other angles to prove that AHTCHA5D70C2.5M crystal is a triclinic crystal system with Figure5 together?

2.It is necessary to have data on the ability of AZLB-Na CHA to remove NH4+ from wastewater and compare with modified chabazite, in order to clearly obtain that whether this method could enhance its ammonium removal.

3.In line 132 and line 383, which solution was used in the regeneration process is contradictory.

4.How about stability of modified chabazite? Whether the crystal structure has changed after ammonia removal.

5.In the study of 3 cycles of modified chabazite, its hard to understand why have shown no loss of three performance because lack of specific data to support.How about recovery capabilities of modified chabazite after 3 cycles?

6.AHTCHA5D70C2.5M in line 97and line 238 should be consistent with the entire article.

Comments on the Quality of English Language

There is no obvious obvious English expression issues in the article.

Reviewer 3 Report

Comments and Suggestions for Authors

In this manuscript, alkaline hydrothermal processes can transform the structure of the parent zeolite into different types. A natural zeolite, was subjected to alkaline hydrothermal treatments with different alkali concentrations, temperatures, and time duration. The manuscript is well written and organized. The results are promising. Some comments should be addressed before any further recommendations:

1- It is more suitable and more informative to express the pollutant concentration in mg/L rather than meq/L.

2- The breakthrough curve has a different shape compared to traditional one. This needs explanation.

3. Some recommendations for future work can be added after the conclusions.

Reviewer 4 Report

Comments and Suggestions for Authors

Natural zeolites are in demand as catalysts in petrochemicals, as feed additives in livestock farming, as adsorbents for wastewater treatment, etc. However, often the properties of these natural materials do not reach the required parameters, so their modification is required. This manuscript discusses an approach to increasing the sorption ammonium capacity of zeolite. Since one of the objectives of this work is to optimize the conditions for the synthesis of zeolite with improved sorption properties, the results of this work correspond to the topics of the Journal «Processes». However, the manuscript needs to be revised and cannot be published without correcting the text.

Comments

1. Authors should carefully read the «Instructions for Authors» for this Journal and prepare the manuscript taking them into account. Pay special attention to the rules for formatting references!

2. In the introduction, the authors write that «By 1962, synthetic zeolites had entered the automobile industry as cracking catalysts». This is true for the petrochemical industry, but not for the automobile industry. Please correct the text.

3. At the end of the Introduction, it is necessary to clearly state the purpose of the work, and not describe its results. Remove the last paragraph describing your results in the Introduction.

4. Figure 3b: Please remove label (b) from the graph.

5. Place the designation of equation 1 on the same line with this equation.

6. Figure 6: The graph does not contain the symbols (! and *) indicated in the text.

7. Figure 8: The appearance of the graph indicates that ammonium adsorption corresponds not to the Langmuir model, but to the Henry model. Therefore, it is necessary to analyze adsorption data using at least three types of Henry, Frendlich and Langmuir equations. And if adsorption is according to the Langmuir model, then what is the significance of the monolayer capacity?

8. Conclusions: It is necessary to indicate not only the practical significance of the work, but also the main results of the research. Give specific values of the obtained parameters.

In general, the manuscript is of great practical importance, therefore, subject to appropriate modification, it can be published in the Journal «Processes». I hope that my comments will be useful to the authors.

Comments on the Quality of English Language

 Minor editing of English language required.

Round 2

Reviewer 2 Report

Comments and Suggestions for Authors

The revised article has more complete content and better logic,which reach the publication requirements.